# LINK PREDICTION WITHOUT GRAPH NEURAL NETWORKS

## ABSTRACT

Link prediction, which consists of predicting edges based on graph features, is a fundamental task in many graph applications. As for several related problems, Graph Neural Networks (GNNs), which are based on an attribute-centric message-passing paradigm, have become the predominant framework for link prediction. GNNs have consistently outperformed traditional topology-based heuristics, but what contributes to their performance? Are there simpler approaches that achieve comparable or better results? To answer these questions, we first identify important limitations in how GNN-based link prediction methods handle the intrinsic class imbalance of the problem—due to the graph sparsity—in their training and evaluation. Moreover, we propose *Gelato*, a novel topology-centric framework that applies a topological heuristic to a graph enhanced by attribute information via graph learning. Our model is trained end-to-end with an N-pair loss on an unbiased training set to address class imbalance. Experiments show that Gelato is 145% more accurate, trains 11 times faster, infers 6,000 times faster, and has less than half of the trainable parameters compared to state-of-the-art GNNs for link prediction.

## 1 INTRODUCTION

Machine learning on graphs supports various structured-data applications including social network analysis (Tang et al., 2008; Li et al., 2017; Qiu et al., 2018b), recommender systems (Jamali & Ester, 2009; Monti et al., 2017; Wang et al., 2019a), natural language processing (Sun et al., 2018a; Sahu et al., 2019; Yao et al., 2019), and physics modeling (Sanchez-Gonzalez et al., 2018; Ivanovic & Pavone, 2019; da Silva et al., 2020). Among the graph-related tasks, one could argue that link prediction (Lü & Zhou, 2011; Martínez et al., 2016) is the most fundamental one. This is because link prediction not only has many concrete applications (Qi et al., 2006; Liben-Nowell & Kleinberg, 2007; Koren et al., 2009) but can also be considered an (implicit or explicit) step of the graph-based machine learning pipeline (Martin et al., 2016; Bahulkar et al., 2018; Wilder et al., 2019)—as the observed graph is usually noisy and/or incomplete.

In recent years, Graph Neural Networks (GNNs) (Kipf & Welling, 2017; Hamilton et al., 2017; Veličković et al., 2018) have emerged as the predominant paradigm for machine learning on graphs. Similar to their great success in node classification (Klicpera et al., 2018; Wu et al., 2019; Zheng et al., 2020) and graph classification (Ying et al., 2018; Zhang et al., 2018a; Morris et al., 2019), GNNs have been shown to achieve state-of-the-art link prediction performance (Zhang & Chen, 2018; Yun et al., 2021; Pan et al., 2022). Compared to classical approaches that rely on expert-designed heuristics to extract topological information (e.g., Common Neighbors (Newman, 2001), Adamic-Adar (Adamic & Adar, 2003), Preferential Attachment (Barabási et al., 2002)), GNNs have the potential to discover new heuristics via supervised learning and the natural advantage of incorporating node attributes.

However, there is little understanding of what factors contribute to the success of GNNs in link prediction, and whether simpler alternatives can achieve comparable performance—as recently found for node classification (Huang et al., 2020). GNN-based methods approach link prediction as a binary classification problem. Yet different from other classification problems, link prediction deals with extremely class-imbalanced data due to the sparsity of real-world graphs. We argue that class imbalance should be accounted for in both training and evaluation of link prediction. In addition, GNNs combine topological and attribute information by learning topology-smoothed attributes (embeddings) via message-passing (Li et al., 2018). This attribute-centric mechanism has been proven

Figure 1: GNN incorporates topology into attributes via message-passing, which is effective for tasks **on** the topology. Link prediction, however, is a task **for** the topology, which motivates the design of Gelato—a novel framework that leverages graph learning to incorporate attributes into topology.

effective for tasks *on* the topology such as node classification (Ma et al., 2020), but link prediction is a task *for* the topology, which naturally motivates topology-centric paradigms (see Figure 1).

The goal of this paper is to address the key issues raised above. We first show that the evaluation of GNN-based link prediction pictures an overly optimistic view of model performance compared to the (more realistic) imbalanced setting. Class imbalance also prevents the generalization of these models due to bias in their training. Instead, we propose the use of the N-pair loss with an unbiased set of training edges to account for class imbalance. Moreover, we present *Gelato*, a novel framework that combines topological and attribute information for link prediction. As a simpler alternative to GNNs, our model applies topology-centric graph learning to incorporate node attributes directly into the graph structure, which is given as input to a topological heuristic, Autocovariance, for link prediction. Extensive experiments demonstrate that our model significantly outperforms state-of-the-art GNN-based methods in both accuracy and scalability.

To summarize, our contributions are: (1) we scrutinize the training and evaluation of supervised link prediction methods and identify their limitations in handling class imbalance; (2) we propose a simple, effective, and efficient framework to combine topological and attribute information for link prediction without using GNNs; and (3) we introduce an N-pair link prediction loss combined with an unbiased set of training edges that we show to be more effective at addressing class imbalance.

## 2 LIMITATIONS IN SUPERVISED LINK PREDICTION EVALUATION & TRAINING

Supervised link prediction is often formulated as a binary classification problem, where the positive (or negative) class includes node pairs connected (or not connected) by a link. A key difference between link prediction and typical classification problems (e.g., node classification) is that the two classes in link prediction are *extremely* imbalanced, since most real-world graphs of interest are sparse (see Table 1). However, we find that class imbalance is not properly addressed in both evaluation and training of existing supervised link prediction approaches, as discussed below.

**Link prediction evaluation.** Area Under the Receiver Operating Characteristic Curve (AUC) and Average Precision (AP) are the two most popular evaluation metrics for supervised link prediction (Kipf & Welling, 2016; Zhang & Chen, 2018; Chami et al., 2019; Zhang et al., 2021; Cai et al., 2021; Yan et al., 2021; Zhu et al., 2021; Chen et al., 2022; Pan et al., 2022). We first argue that, as in other imbalanced classification problems (Saito & Rehmsmeier, 2015), AUC is not an effective evaluation metric for link prediction as it is biased towards the majority class (non-edges). On the other hand, AP and other rank-based metrics such as Hits@$k$—used in Open Graph Benchmark (OGB) (Hu et al., 2020)—are effective for imbalanced classification *if evaluated on a test set that follows the original class distribution*. Yet, existing link prediction methods (Kipf & Welling, 2016; Zhang & Chen, 2018; Cai et al., 2021; Zhu et al., 2021; Pan et al., 2022) compute AP on a test set that contains all positive test pairs and only an equal number of random negative pairs. Similarly, OGB computes Hits@$k$ against a very small subset of random negative pairs. We term these approaches *biased testing* as they highly overestimate the ratio of positive pairs in the graph. Evaluation metrics based on these biased test sets provide an overly optimistic measurement of the actual performance in *unbiased testing*, where every negative pair is included in the test set. In fact, in real applications where test positive edges are not known a priori, it is impossible to construct those biased test sets to begin with. Below, we also present an illustrative example of the misleading performance evaluation based on *biased testing*.

**Example**: Consider a graph with 10k nodes, 100k edges, and 99.9M disconnected (or negative) pairs. A (bad) model that ranks 1M false positives higher than the true edges achieves 0.99 AUC and 0.95 in AP under *biased testing* with equal negative samples. (Detailed computation in Appendix A.)

The above discussion motivates a more representative evaluation setting for supervised link prediction. Specifically, we argue for the use of rank-based evaluation metrics—AP, Precision@$k$ (Lü & Zhou, 2011), and Hits@$k$ (Bordes et al., 2013)—with *unbiased testing*, where positive edges are ranked against all negative pairs. These metrics have been widely applied in related problems, such as unsupervised link prediction (Lü & Zhou, 2011; Ou et al., 2016; Zhang et al., 2018b; Huang et al., 2021), knowledge graph completion (Bordes et al., 2013; Yang et al., 2015; Sun et al., 2018b), and information retrieval (Schütze et al., 2008), where class imbalance is also significant. In our experiments, we will illustrate how these evaluation metrics combined with *unbiased testing* provide a drastically different and more informative performance evaluation compared to existing approaches.

**Link prediction training.** Following the formulation of supervised link prediction as binary classification, most existing models adopt the binary cross entropy loss to optimize their parameters (Kipf & Welling, 2016; Zhang & Chen, 2018; Chami et al., 2019; Zhang et al., 2021; Yan et al., 2021; Yun et al., 2021; Zhu et al., 2021; Chen et al., 2022). To deal with class imbalance, these approaches downsample the negative pairs to match the number of positive pairs in the training set (*biased training*). We highlight two drawbacks of *biased training*: (1) it induces the model to overestimate the probability of positive pairs, and (2) it discards potentially useful evidence from most negative pairs. Notice that the first drawback is often hidden by *biased testing*. Instead, this paper proposes the use of *unbiased training*, where the ratio of negative pairs in the training set is the same as in the input graph. To train our model in this highly imbalanced setting, we apply the N-pair loss for link prediction instead of the cross entropy loss (Section 3.3).

## 3 METHOD

**Notation and problem.** Consider an attributed graph $G = (V, E, X)$, where $V$ is the set of $n$ nodes, $E$ is the set of $m$ edges (links), and $X = (x_1, ..., x_n)^T \in \mathbb{R}^{n \times r}$ collects $r$-dimensional node attributes. The topological (structural) information of the graph is represented by its adjacency matrix $A \in \mathbb{R}^{n \times n}$, with $A_{uv} > 0$ if an edge of weight $A_{uv}$ connects nodes $u$ and $v$ and $A_{uv} = 0$, otherwise. The (weighted) degree of node $u$ is given as $d_u = \sum_v A_{uv}$ and the corresponding degree vector (matrix) is denoted as $d \in \mathbb{R}^n$ ($D \in \mathbb{R}^{n \times n}$). The volume of the graph is $\text{vol}(G) = \sum_u d_u$. Our goal is to infer missing links in $G$ based on its topological and attribute information, $A$ and $X$.

**Model overview.** Figure 2 provides an overview of our link prediction model. It starts with a topology-centric graph learning phase that incorporates node attribute information directly into the graph structure via a Multi-layer Perceptron (MLP). We then apply a topological heuristic, Autocovariance (AC), to the attribute-enhanced graph to obtain a pairwise score matrix. Node pairs with the highest scores are predicted as (positive) links. The scores for training pairs are collected to compute an N-pair loss. Finally, the loss is used to train the MLP parameters in an end-to-end manner. We named our model Gelato (Graph enhancement for link prediction with autocovariance). Gelato represents a paradigm shift in supervised link prediction by combining a graph encoding of attributes with a topological heuristic instead of relying on increasingly popular GNN-based embeddings.

### 3.1 GRAPH LEARNING

The goal of graph learning is to generate an enhanced graph that incorporates node attribute information into the topology. This can be considered as the "dual" operation of message-passing in GNNs, which incorporates topological information into attributes (embeddings). We argue that graph learning is the more suitable scheme to combine attributes and topology for link prediction, since link prediction is a task for the topology itself (as opposed to other applications such as node classification).

Specifically, our first step of graph learning is to augment the original edges with a set of node pairs based on their (untrained) attribute similarity (i.e., adding an $\epsilon$-neighborhood graph):

$$\widetilde{E} = E + \{(u, v) \mid s(x_u, x_v) > \epsilon_\eta\} \tag{1}$$

where $s(\cdot)$ can be any similarity function (we use cosine in our experiments) and $\epsilon_\eta$ is a threshold that determines the number of added pairs as a ratio $\eta$ of the original number of edges $m$.

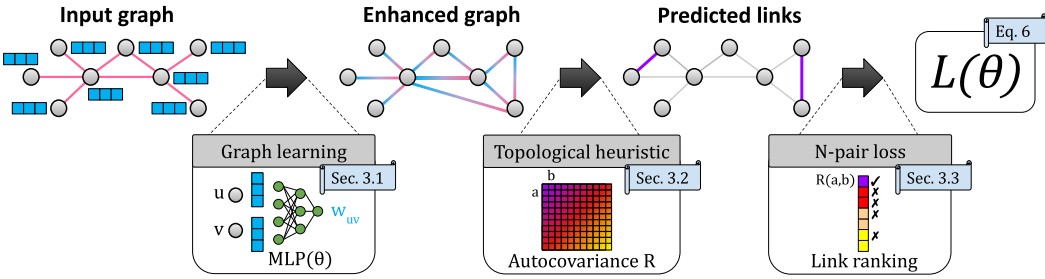

Figure 2: Gelato applies graph learning to incorporate attribute information into the topology via an MLP. The learned graph is given to a topological heuristic that predicts edges between node pairs with high Autocovariance similarity. The parameters of the MLP are optimized end-to-end using the N-pair loss. Experiments show that Gelato outperforms state-of-the-art GNN-based link prediction methods.

A simple MLP then maps the pairwise node attributes into a trained edge weight for every edge in $\widetilde{E}$:

$$w_{uv} = \text{MLP}([x_u; x_v]; \theta) \tag{2}$$

where $[x_u; x_v]$ denotes the concatenation of $x_u$ and $x_v$ and $\theta$ contains the trainable parameters. For undirected graphs, we instead use the following permutation invariant operator (Chen et al., 2014):

$$w_{uv} = \text{MLP}([x_u + x_v; |x_u - x_v|]; \theta) \tag{3}$$

The final edge weights of the enhanced graph are a weighted combination of the topological weights, the untrained weights, and the trained weights:

$$\widetilde{A}_{uv} = \alpha A_{uv} + (1 - \alpha)(\beta w_{uv} + (1 - \beta)s(x_u, x_v)) \tag{4}$$

where $\alpha$ and $\beta$ are hyperparameters. The enhanced adjacency matrix $\widetilde{A}$ is then fed into a topological heuristic for link prediction introduced in the next section. Note that the MLP is not trained directly to predict the links, but instead trained end-to-end to enhance the input graph given to the topological heuristic. Also note that the MLP can be easily replaced by a more powerful model such as a GNN, but the goal of this paper is to demonstrate the general effectiveness of our framework and we will show that even a simple MLP leads to significant improvement over the base heuristic.

## 3.2 TOPOLOGICAL HEURISTIC

Assuming that the learned adjacency matrix $\widetilde{A}$ incorporates both structural and attribute information, Gelato applies a topological heuristic to $\widetilde{A}$. Specifically, we adopt Autocovariance, which has been shown to achieve state-of-the-art link prediction results for non-attributed graphs (Huang et al., 2021).

Autocovariance is a random-walk based similarity metric. Intuitively, it measures the difference between the co-visiting probabilities for a pair of nodes in a truncated walk and in an infinitely long walk. Given the enhanced graph $\widetilde{G}$, the Autocovariance similarity matrix $R \in \mathbb{R}^{n \times n}$ is given as

$$R = \frac{\widetilde{D}}{\text{vol}(\widetilde{G})}(\widetilde{D}^{-1}\widetilde{A})^t - \frac{\tilde{d}\tilde{d}^T}{\text{vol}^2(\widetilde{G})} \tag{5}$$

where $t \in \mathbb{N}_0$ is the scaling parameter of the truncated walk. Each entry $R_{uv}$ represents a similarity score for node pair $(u, v)$ and top similarity pairs are predicted as links. Note that $R_{uv}$ only depends on the $t$-hop enclosing subgraph of $(u, v)$ and can be easily differentiated with respect to the edge weights in the subgraph. In fact, Gelato could be applied with any differentiable topological heuristic or even a combination of them. In our experiments (Section 4.2), we will show that Autocovariance alone enables state-of-the-art link prediction performance.

Next, we introduce how to train our model parameters with supervised information.

## 3.3 N-PAIR LOSS AND UNBIASED TRAINING

As we have mentioned in Section 2, current supervised link prediction methods rely on *biased training* and the cross entropy loss (CE) to optimize model parameters. Instead, Gelato applies the N-pair

loss (Sohn, 2016) that is inspired by the metric learning and learning-to-rank literature (McFee & Lanckriet, 2010; Cakir et al., 2019; Revaud et al., 2019; Wang et al., 2019b) to train the parameters of our graph learning model (see Section 3.1) from highly imbalanced *unbiased training* data.

The N-pair loss (NP) contrasts each positive training edge $(u, v)$ against a set of negative pairs $N(u, v)$. It is computed as follows:

$$L(\theta) = - \sum_{(u,v) \in E} \log \frac{\exp(R_{uv})}{\exp(R_{uv}) + \sum_{(p,q) \in N(u,v)} \exp(R_{pq})} \tag{6}$$

Intuitively, $L(\theta)$ is minimized when each positive edge $(u, v)$ has a much higher similarity than its contrasted negative pairs: $R_{uv} \gg R_{pq}, \forall (p, q) \in N(u, v)$. Compared to CE, NP is more sensitive to negative pairs that have comparable similarities to those of positive pairs—they are more likely to be false positives. While NP achieves good performance in our experiments, alternative losses from the learning-to-rank literature (Freund et al., 2003; Xia et al., 2008; Bruch, 2021) could also be applied.

Gelato generates negative samples $N(u, v)$ using *unbiased training*. This means that $N(u, v)$ is a random subset of all disconnected pairs in the training graph, and $|N(u, v)|$ is proportional to the ratio of negative pairs over positive ones. In this way, we leverage more information contained in negative pairs compared to *biased training*. Note that, similar to *unbiased training*, (unsupervised) topological heuristics implicitly use information from all edges and non-edges. Also, *unbiased training* can be combined with adversarial negative sampling methods (Cai & Wang, 2018; Wang et al., 2018) from the knowledge graph embedding literature to increase the quality of contrasted negative pairs.

**Complexity analysis.** The only trainable component in our model is the graph learning MLP with $O(rh + lh^2)$ parameters—where $r$ is the number of node features, $l$ is the number of hidden layers, and $h$ is the number of neurons per layer. Notice that the number of parameters is independent of the graph size. Constructing the $\epsilon$-neighborhood graph based on cosine similarity can be done efficiently using hashing and pruning (Satuluri & Parthasarathy, 2012; Anastasiu & Karypis, 2014). Computing the enhanced adjacency matrix with the MLP takes $O((1+\eta)mr)$ time per epoch—where $m = |E|$ and $\eta$ is the ratio of edges added to $E$ from the $\epsilon$-neighborhood graph. We apply sparse matrix multiplication to compute entries of the $t$-step AC in $O((1 + \eta)mt)$ time. Note that unlike recent GNN-based approaches (Zhang & Chen, 2018; Liu et al., 2020; Pan et al., 2022) that generate distinctive subgraphs for each link (e.g., via the labeling trick), enclosing subgraphs for links in Gelato share the same information (i.e., learned edge weights), which significantly reduces the computational cost. Our experiments will demonstrate Gelato's efficiency in training and inference.

## 4 EXPERIMENTS

We provide empirical evidence for our claims regarding supervised link prediction and demonstrate the accuracy and efficiency of Gelato. Our implementation is anonymously available at `https://anonymous.4open.science/r/Gelato/`.

### 4.1 EXPERIMENT SETTINGS

**Datasets.** Our method is evaluated on five attributed graphs commonly used as link prediction benchmark (Chami et al., 2019; Zhang et al., 2021; Yan et al., 2021; Zhu et al., 2021; Chen et al., 2022; Pan et al., 2022). Table 1 shows dataset statistics—see Appendix B for dataset details.

Table 1: A summary of dataset statistics.

|  |  | #Nodes | #Edges | #Attributes | Avg. degree | Density |
|---|---|---|---|---|---|---|
| | CORA | 2,708 | 5,278 | 1,433 | 3.90 | 0.14% |
| Yang et al. (2016) | CITESEER | 3,327 | 4,552 | 3,703 | 2.74 | 0.08% |
| | PUBMED | 19,717 | 44,324 | 500 | 4.50 | 0.02% |
| Shchur et al. (2018) | PHOTO | 7,650 | 119,081 | 745 | 31.13 | 0.41% |
| | COMPUTERS | 13,752 | 245,861 | 767 | 35.76 | 0.26% |

**Baselines.** For GNN-based link prediction, we include six state-of-the-art methods published in the past two years: LGCN (Zhang et al., 2021), TLC-GNN (Yan et al., 2021), Neo-GNN (Yun et al.,

2021), NBFNet (Zhu et al., 2021), BScNets (Chen et al., 2022), and WalkPool (Pan et al., 2022), as well as three pioneering works—GAE (Kipf & Welling, 2016), SEAL (Zhang & Chen, 2018), and HGCN (Chami et al., 2019). For topological link prediction heuristics, we consider Common Neighbors (CN) (Newman, 2001), Adamic Adar (AA) (Adamic & Adar, 2003), Resource Allocation (RA) (Zhou et al., 2009), and Autocovariance (AC) (Huang et al., 2021)—the base heuristic in our model. To demonstrate the superiority of the proposed end-to-end model, we also include an MLP trained directly for link prediction, the cosine similarity (Cos) between node attributes, and AC on top of the respective weighted/augmented graphs (i.e., two-stage approaches) as baselines.

**Hyperparameters.** For Gelato, we tune the proportion of added edges $\eta$ from $\{0.0, 0.25, 0.5, 0.75, 1.0\}$, the topological weight $\alpha$ from $\{0.0, 0.25, 0.5, 0.75\}$, and the trained weight $\beta$ from $\{0.25, 0.5, 0.75, 1.0\}$, with a sensitivity analysis included in Appendix C. All other settings are fixed across datasets: MLP with one hidden layer of 128 neurons, AC scaling parameter $t = 3$, Adam optimizer (Kingma & Ba, 2015) with a learning rate of 0.001, a dropout rate of 0.5, and *unbiased training* without downsampling. For baselines, we use the same hyperparameters as in their papers.

**Link prediction setting.** Following Kipf & Welling (2016); Zhang & Chen (2018); Chami et al. (2019); Zhang et al. (2021); Chen et al. (2022); Pan et al. (2022), we randomly split edges into 85%/5%/10% for training, validation, and testing. As argued in Section 2, to better reflect the performance in real-world applications, we adopt *unbiased testing* and use Precision@$k$ ($prec@k$)—proportion of positive edges among the top $k$ of all testing pairs, Hits@$k$ ($hits@k$)—ratio of positive edges individually ranked above $k$th place against all negative pairs, and Average Precision (AP)—area under the precision-recall curve, as evaluation metrics. We report results from 10 runs with random seeds ranging from 1 to 10. More detailed experiment settings can be found in Appendix D.

## 4.2 LINK PREDICTION PERFORMANCE

Table 2 summarizes the link prediction performance in terms of the mean and standard deviation of AP of all methods. Figure 3 and Figure 4 show results based on $prec@k$ and $hits@k$ for varying $k$ values.

Table 2: Link prediction performance comparison (mean ± std AP). Gelato consistently outperforms GNN-based methods, topological heuristics, and two-stage approaches combining attributes/topology.

| | | CORA | CITESEER | PUBMED | PHOTO | COMPUTERS |
|---|---|---|---|---|---|---|
| GNN | GAE | 0.27 ± 0.02 | 0.66 ± 0.11 | 0.26 ± 0.03 | 0.28 ± 0.02 | 0.30 ± 0.02 |
| | SEAL | 1.89 ± 0.74 | 0.91 ± 0.66 | *** | 10.49 ± 0.86 | 6.84[*] |
| | HGCN | 0.82 ± 0.03 | 0.74 ± 0.10 | 0.35 ± 0.01 | 2.11 ± 0.10 | 2.30 ± 0.14 |
| | LGCN | 1.14 ± 0.04 | 0.86 ± 0.09 | 0.44 ± 0.01 | 3.53 ± 0.05 | 1.96 ± 0.03 |
| | TLC-GNN | 0.29 ± 0.09 | 0.35 ± 0.18 | OOM | 1.77 ± 0.11 | OOM |
| | Neo-GNN | 2.05 ± 0.61 | 1.61 ± 0.36 | 1.21 ± 0.14 | 10.83 ± 1.53 | 6.75[*] |
| | NBFNet | 1.36 ± 0.17 | 0.77 ± 0.22 | *** | 11.99 ± 1.60 | *** |
| | BScNets | 0.32 ± 0.08 | 0.20 ± 0.06 | 0.22 ± 0.08 | 2.47 ± 0.18 | 1.45 ± 0.10 |
| | WalkPool | 2.04 ± 0.07 | 1.39 ± 0.11 | 1.31[*] | OOM | OOM |
| Topological Heuristics | CN | 1.10 ± 0.00 | 0.74 ± 0.00 | 0.36 ± 0.00 | 7.73 ± 0.00 | 5.09 ± 0.00 |
| | AA | 2.07 ± 0.00 | 1.24 ± 0.00 | 0.45 ± 0.00 | 9.67 ± 0.00 | 6.52 ± 0.00 |
| | RA | 2.02 ± 0.00 | 1.19 ± 0.00 | 0.33 ± 0.00 | 10.77 ± 0.00 | 7.71 ± 0.00 |
| | AC | 2.43 ± 0.00 | 2.65 ± 0.00 | 2.50 ± 0.00 | 16.63 ± 0.00 | 11.64 ± 0.00 |
| Attributes + Topology | MLP | 0.30 ± 0.05 | 0.44 ± 0.09 | 0.14 ± 0.06 | 1.01 ± 0.26 | 0.41 ± 0.23 |
| | Cos | 0.42 ± 0.00 | 1.89 ± 0.00 | 0.07 ± 0.00 | 0.11 ± 0.00 | 0.07 ± 0.00 |
| | MLP+AC | 3.24 ± 0.03 | 1.95 ± 0.05 | 2.61 ± 0.06 | 15.99 ± 0.21 | 11.25 ± 0.13 |
| | Cos+AC | 3.60 ± 0.00 | 4.46 ± 0.00 | 0.51 ± 0.00 | 10.01 ± 0.00 | 5.20 ± 0.00 |
| | MLP+Cos+AC | 3.39 ± 0.06 | 4.15 ± 0.14 | 0.55 ± 0.03 | 10.88 ± 0.09 | 5.75 ± 0.11 |
| Gelato | | **3.90 ± 0.03** | **4.55 ± 0.02** | **2.88 ± 0.09** | **25.68 ± 0.53** | **18.77 ± 0.19** |

[*] Run only once as each run takes ~100 hrs;   *** Each run takes >1000 hrs;   OOM: Out Of Memory.

First, we want to highlight the drastically different performance of GNN-based methods compared to those found in the original papers (Zhang et al., 2021; Yan et al., 2021; Yun et al., 2021; Zhu et al., 2021; Chen et al., 2022; Pan et al., 2022) and reproduced in Appendix E. While they achieve AUC/AP scores of often higher than 90% under *biased testing*, here we see most of them underperform even

the simplest topological heuristics such as Common Neighbors under *unbiased testing*. These results support our arguments from Section 2 that evaluation metrics based on *biased testing* can produce misleading results compared to *unbiased testing*. The overall best performing GNN model is Neo-GNN, which directly generalizes the pairwise topological heuristics. Yet still, it consistently underperforms AC, a random-walk based heuristic that needs neither node attributes nor supervision/training.

We then look at two-stage combinations of AC and models for attribute information. We observe noticeable performance gains from combining attribute cosine similarity and AC in CORA and CITE-SEER but not for other datasets. Other two-stage approaches achieve similar or worse performance.

Finally, Gelato significantly outperforms the best GNN-based model with an average relative gain of **145.2%** and AC with a gain of **52.6%** in terms of AP—similar results were obtained for $prec@k$ and $hits@k$. This validates our hypothesis that a simple MLP can effectively incorporate node attribute information into the topology when our model is trained end-to-end. Next, we will provide insights behind these improvements and demonstrate the efficiency of Gelato on training and inference.

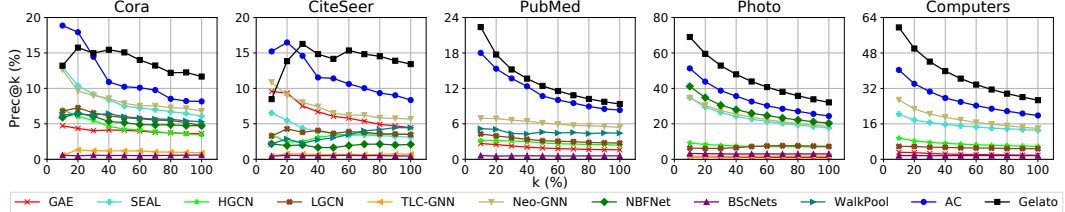

Figure 3: Link prediction performance in terms of $prec@k$ for varying values of $k$ (as percentages of test edges). With few exceptions, Gelato outperforms the baselines across different values of $k$.

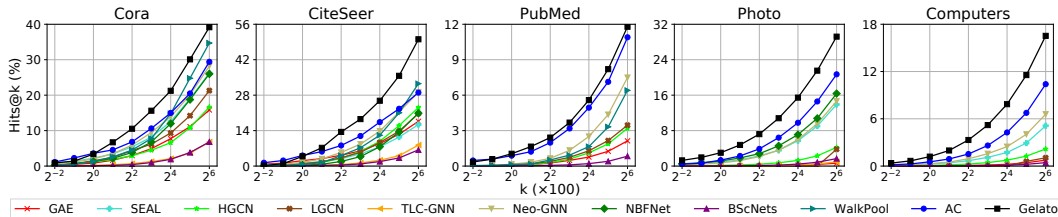

Figure 4: Link prediction performance in terms of $hits@k$ for varying values of $k$. With few exceptions, Gelato outperforms the baselines across different values of $k$.

### 4.3 VISUALIZING GELATO LINK PREDICTION PROCESS

To better understand the performance of Gelato, we visualize the input adjacency matrix and node attributes (in terms of pairwise Euclidean distance), the attribute-enhanced adjacency matrix, and the final prediction scores in Figure 5. The results are based on a subgraph of PHOTO containing the top 160 nodes belonging to the first class sorted in decreasing order of their (within-class) degree.

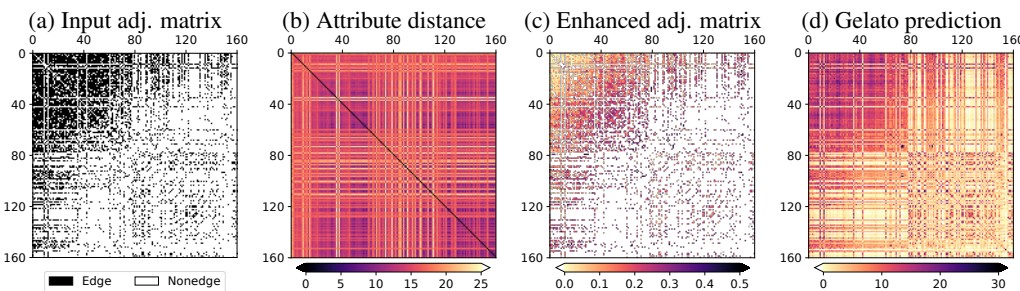

Figure 5: Illustration of the link prediction process of Gelato. Graph learning effectively incorporates node attributes into topology and AC on the enhanced graph enables state-of-the-art link prediction.

We observe that the graph learning module of Gelato generates a more informative adjacency matrix (Figure 5c) by incorporating node attributes (Figure 5b) into the input adjacency matrix (Figure 5a). This can be seen from the down-weighting of the edges connecting the high-degree nodes with larger attribute distances (matrix entries 0-40 and especially 0-10) and the up-weighting of those connecting medium-degree nodes with smaller distances (40-80). Applying the topological heuristic, Autocovariance, to this enhanced adjacency matrix thus leverages the advantages from both worlds (Figure 5d). It not only covers most true edges between high-degree nodes as AC captures node degree distributions (Huang et al., 2021) but also avoids false predictions for connections between high-degree and low-degree nodes thanks to the attribute-based edge down-weighting, enabling state-of-the-art link prediction performance for attributed graphs. A detailed analysis of the improvements of Gelato over the vanilla AC and comparisons with the GNN-based link prediction process is included in Appendix F.

## 4.4 LOSS AND TRAINING SETTING

In this section, we demonstrate the advantages of the proposed N-pair loss and *unbiased training* for supervised link prediction. Figure 6 shows the training and validation losses and $prec@100\%$ (our validation metric) in training Gelato based on the cross entropy (CE) and N-pair (NP) losses under *biased* and *unbiased training* respectively. The final test AP scores are shown in the titles.

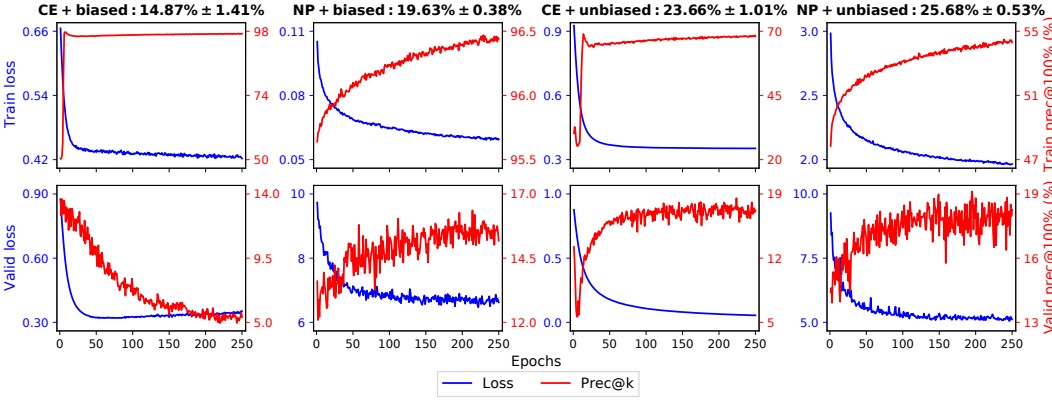

Figure 6: Training of Gelato based on different losses and training settings for PHOTO with test AP (mean ± std) shown in the titles. Compared with the cross entropy loss, the N-pair loss with *unbiased training* is a more consistent proxy for *unbiased testing* metrics and leads to better peak performance.

In the first column (CE with *biased training*), different from the training, both loss and precision for (unbiased) validation decrease. This leads to even worse test performance compared to the untrained base model (i.e., AC). In other words, albeit being the most popular loss function for supervised link prediction, CE with *biased training* does not generalize to *unbiased testing*. On the contrary, as shown in the second column, the proposed NP loss with *biased training*—equivalent to the pairwise logistic loss (Burges et al., 2005)—is a more effective proxy for *unbiased testing* metrics.

The right two columns show results with *unbiased training*, which is better for CE as more negative pairs are present in the training set (with the original class ratio). On the other hand, NP is more consistent with unbiased evaluation metrics, leading to 8.5% better performance. This is because, unlike CE, which optimizes positive and negative pairs independently, NP contrasts negative pairs against positive ones, giving higher importance to negative pairs that are more likely to be false positives.

An ablation study of Gelato is included in Appendix G. While all supervised baselines originally adopt *biased training*, we also implement the same *unbiased training* (and N-pair loss) as Gelato for those that are scalable in Appendix H—results are consistent with the ones discussed in Section 4.2.

## 4.5 RUNNING TIME

We compare Gelato with other supervised link prediction methods in terms of running time for PHOTO in Table 3. As the only method that applies *unbiased training*—with more negative samples—Gelato

shows a competitive training speed that is $11\times$ faster than the best performing GNN-based method, Neo-GNN. In terms of inference time, Gelato is much faster than most baselines with a $6{,}000\times$ speedup compared to Neo-GNN. We further observe more significant efficiency gains for Gelato over Neo-GNN for larger datasets—e.g., $14\times$ (training) and $25{,}000\times$ (testing) for COMPUTERS.

Table 3: Training and inference time comparison between supervised link prediction methods for PHOTO. Gelato has competitive training time (even under *unbiased training*) and is significantly faster than most baselines in terms of inference, especially compared to the best GNN model, Neo-GNN.

|  | GAE | SEAL | HGCN | LGCN | TLC-GNN | Neo-GNN | NBFNet | BScNets | MLP | Gelato |
|---|---|---|---|---|---|---|---|---|---|---|
| Training | 1,022 | 11,493 | 92 | 56 | 42,440 | 14,807 | 30,896 | 115 | 232 | 1,265 |
| Testing | 0.031 | 380 | 0.093 | 0.099 | 5.722 | 346 | 76,737 | 0.394 | 1.801 | 0.057 |

## 5 RELATED WORK

**Topological heuristics for link prediction.** The early link prediction literature focuses on topology-based heuristics. This includes approaches based on local (e.g., Common Neighbors (Newman, 2001), Adamic Adar (Adamic & Adar, 2003), and Resource Allocation (Zhou et al., 2009)) and higher-order (e.g., Katz (Katz, 1953), PageRank (Page et al., 1999), and SimRank (Jeh & Widom, 2002)) information. More recently, random-walk based graph embedding methods, which learn vector representations for graph data (Perozzi et al., 2014; Grover & Leskovec, 2016; Huang et al., 2021), have achieved promising results in graph machine learning tasks. Popular embedding approaches, such as DeepWalk (Perozzi et al., 2014) and node2vec (Grover & Leskovec, 2016), have been shown to implicitly approximate the Pointwise Mutual Information similarity (Qiu et al., 2018a), which can also be used as a link prediction heuristic. This has motivated the investigation of alternative similarity metrics such as Autocovariance (Delvenne et al., 2010; Huang et al., 2021; 2022). However, these heuristics are unsupervised and cannot take advantage of data beyond the topology.

**Graph Neural Networks for link prediction.** GNN-based link prediction addresses the limitations of topological heuristics by training a neural network to capture both topological and attribute information and potentially learn new heuristics via supervised learning. GAE (Kipf & Welling, 2016) combines a graph convolution network (Kipf & Welling, 2017) and an inner product decoder based on node embeddings for link prediction. SEAL (Zhang & Chen, 2018) models link prediction as a binary subgraph classification problem (edge/non-edge), and follow-up work (e.g., SHHF (Liu et al., 2020), WalkPool (Pan et al., 2022)) investigates different pooling strategies. Other recent approaches for GNN-based link prediction include learning representations in hyperbolic space (e.g., HGCN (Chami et al., 2019), LGCN (Zhang et al., 2021)), generalizing topological heuristics (e.g., Neo-GNN (Yun et al., 2021), NBFNet (Zhu et al., 2021)), and incorporating additional topological features (e.g., TLC-GNN (Yan et al., 2021), BScNets (Chen et al., 2022)). Motivated by the growing popularity of GNNs for link prediction, this work investigates key questions regarding their training, evaluation, and ability to learn effective topological heuristics directly from data. We propose Gelato, which is simpler, more accurate, and faster than the state-of-the-art GNN-based link prediction methods.

**Graph learning.** Gelato learns a graph that combines topological and attribute information. Graph learning also enables the application of GNNs when the graph is unavailable, noisy, or incomplete. LDS (Franceschi et al., 2019) jointly learns a probability distribution over edges and GNN parameters. IDGL (Chen et al., 2020) and EGLN (Yang et al., 2021) alternate between optimizing the graph and embeddings for node/graph classification and collaborative filtering. Singh et al. (2021) proposes two-stage link prediction by augmenting the graph as a preprocessing step. In comparison, Gelato effectively learns a graph in an end-to-end manner by minimizing the loss of a topological heuristic.

## 6 CONCLUSION

This work sheds light on key limitations in how GNN-based link prediction methods handle the intrinsic class imbalance of the problem and presents more effective and efficient ways to combine attributes and topology. Our findings might open new research directions on machine learning for graph data, including a better understanding of the advantages of increasingly popular and sophisticated deep learning models compared to more traditional and simpler graph-based solutions.

ETHICS STATEMENT

Link prediction might be used to disclose private user information (e.g. in social and communication networks). Further research is needed to better protect such sensitive information and prevent its misuse by governments, corporations, and other possibly ill-intentioned parties.

REPRODUCIBILITY STATEMENT

Our code is anonymously available at `https://anonymous.4open.science/r/Gelato/`, which includes a reference implementation of the proposed model, dataset splits, pretrained model artifacts, tuned hyperparameters, and detailed instruction for reproducing our results. Our main findings— Table 2, Figure 3, and Figure 4—can be reproduced with one line of code using our repository. In addition, a detailed description of the datasets can be found in Appendix B, a detailed discussion on hyperparameter choices can be found in Appendix C, and detailed experiment settings and reference of baseline repositories can be found in Appendix D.

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

# A  ANALYSIS OF LINK PREDICTION EVALUATION METRICS WITH DIFFERENT TEST SETTINGS

In Section 2, we present an example scenario where a bad link prediction model that ranks 1M false positives higher than the 100k true edges achieves good AUC/AP with *biased testing*. Here, we provide the detailed computation steps and compare the results with those based on *unbiased testing*.

Figure 7a and Figure 7b show the receiver operating characteristic (ROC) and precision-recall (PR) curves for the model under *biased testing* with equal negative samples. Due to the downsampling, only 100k (out of 99.9M) negative pairs are included in the test set, among which only $\frac{100k}{99.9M} \times 1M \approx 1k$ pairs are ranked higher than the positive edges. In the ROC curve, this means that once the false positive rate reaches $\frac{1k}{100k} = 0.01$, the true positive rate would reach 1.0, leading to an AUC score of 0.99. Similarly, in the PR curve, when the recall reaches 1.0, the precision is $\frac{100k}{1k+100k} \approx 0.99$, leading to an overall AP score of ~0.95.

By comparison, as shown in Figure 7c, when the recall reaches 1.0, the precision under *unbiased testing* is only $\frac{100k}{1M+100k} \approx 0.09$, leading to an AP score of ~0.05. This demonstrates that evaluation metrics based on *biased testing* provide an overly optimistic measurement of link prediction model performance compared to the more realistic *unbiased testing* setting.

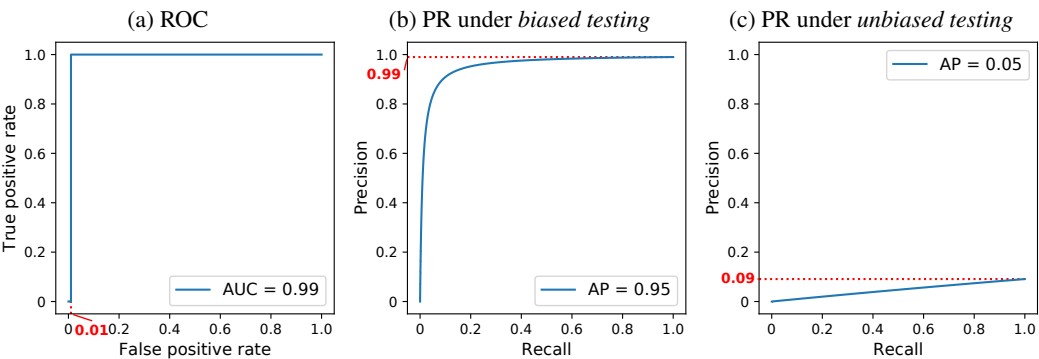

Figure 7: Receiver operating characteristic and precision-recall curves for the bad link prediction model that ranks 1M false positives higher than the 100k true edges. The model achieves 0.99 in AUC and 0.95 AP under *biased testing*, while the more informative performance evaluation metric, Average Precision (AP) under *unbiased testing*, is only 0.05.

# B  DESCRIPTION OF DATASETS

We use the following datasets in our experiments (with their statistics in Table 1):

- CORA (McCallum et al., 2000) and CiteSeer (Giles et al., 1998) are citation networks where nodes represent scientific publications (classified into seven and six classes, respectively) and edges represent the citations between them. Attributes of each node is a binary word vector indicating the absence/presence of the corresponding word from a dictionary.

- PUBMED (Sen et al., 2008) is a citation network where nodes represent scientific publications (classified into three classes) and edges represent the citations between them. Attributes of each node is a TF/IDF weighted word vector.

- PHOTO and COMPUTERS are subgraphs of the Amazon co-purchase graph (McAuley et al., 2015) where nodes represent products (classified into eight and ten classes, respectively) and edges imply that two products are frequently bought together. Attributes of each node is a bag-of-word vector encoding the product review.

We use the publicly available version of the datasets from the `pytorch-geometric` library (Fey & Lenssen, 2019) (under the MIT licence) curated by Yang et al. (2016) and Shchur et al. (2018).

## C SELECTED HYPERPARAMETERS AND SENSITIVITY ANALYSIS

The selected hyperparameters of Gelato for each dataset are recorded in Table 4, and a sensitivity analysis of $\eta$ and $\alpha$ and $\beta$ are shown in Figure 8 and Figure 9 respectively for PHOTO and CORA.

Table 4: Selected hyperparameters of Gelato.

|  | CORA | CITESEER | PUBMED | PHOTO | COMPUTERS |
|---|---|---|---|---|---|
| Proportion of added edges $\eta$ | 0.5 | 0.75 | 0.0 | 0.0 | 0.0 |
| Topological weight $\alpha$ | 0.5 | 0.5 | 0.0 | 0.0 | 0.0 |
| Trained weight $\beta$ | 0.25 | 0.5 | 1.0 | 1.0 | 1.0 |

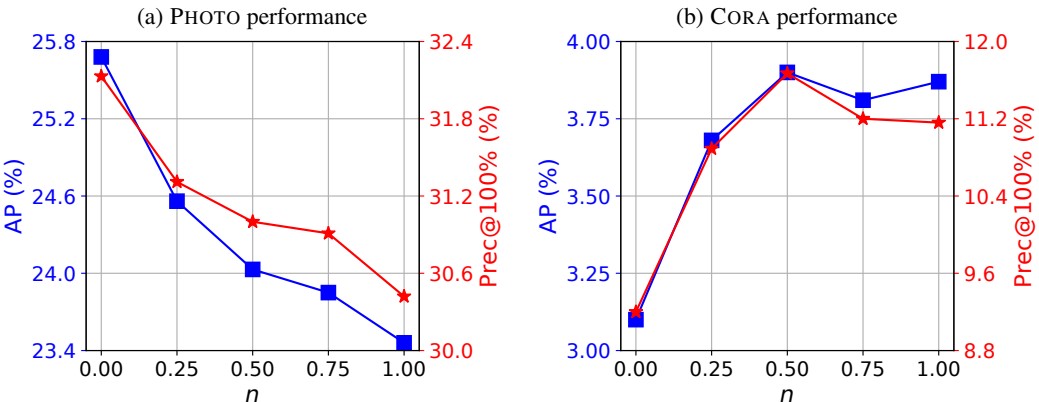

Figure 8: Performance of Gelato with different values of $\eta$.

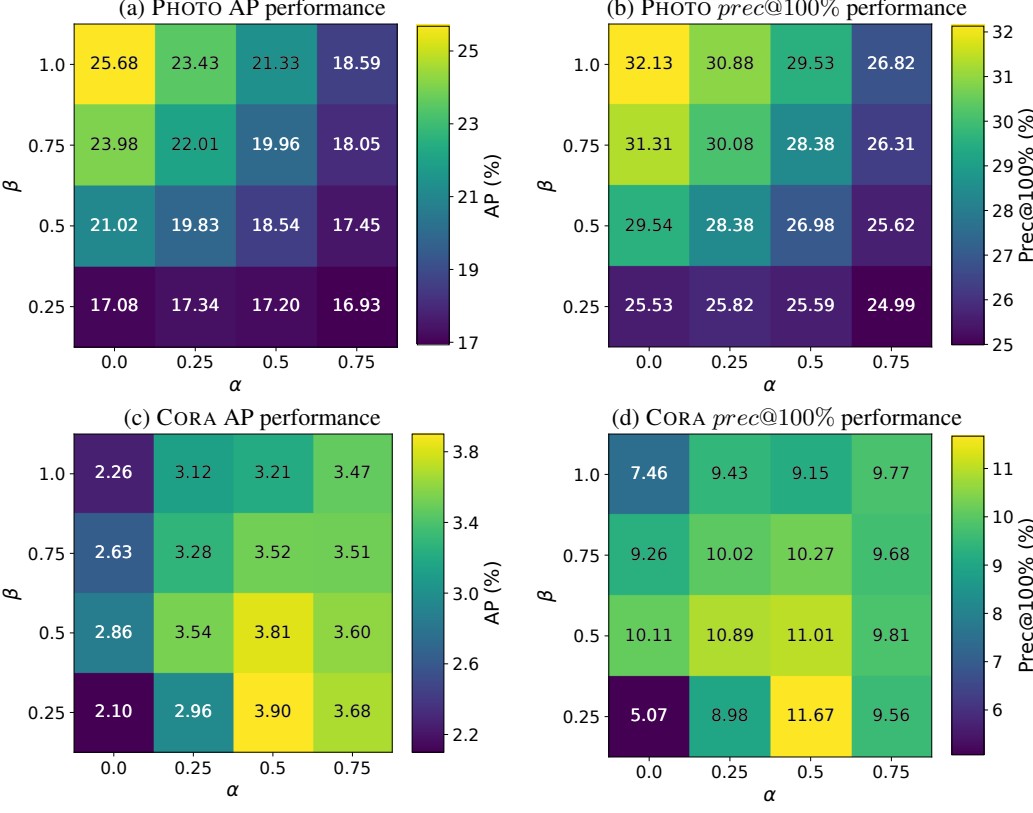

Figure 9: Performance of Gelato with different combinations of $\alpha$ and $\beta$.

For most datasets, we find that simply setting $\beta = 1.0$ and $\eta = \alpha = 0.0$ leads to the best performance, corresponding to the scenario where no edges based on cosine similarity are added and the edge weights are completely learned by the MLP. For CORA and CITESEER, however, we first notice that adding edges based on untrained cosine similarity alone leads to improved performance (see Table 2), which motivates us to set $\eta = 0.5/0.75$. In addition, we find that a large trainable weight $\beta$ leads to overfitting of the model as the number of node attributes is large while the number of (positive) edges is small for CORA and CITESEER (see Table 1). We address this by decreasing the relative importance of trained edge weights ($\beta = 0.25/0.5$) and increasing that of the topological edge weights ($\alpha = 0.5$), which leads to better generality and improved performance. Based on our experiments, these hyperparameters can be easily tuned using simple hyperparameter search techniques, such as line search, using a small validation set.

## D  DETAILED EXPERIMENT SETTINGS

**Data splits for unbiased training and unbiased testing.** We first randomly (with seed 0) split the edges $E$ of the original graph into $E_{train}^{+}$, $E_{valid}^{+}$, and $E_{test}^{+}$ for training, validation, and testing. Then, we set the negative pairs in these three sets as (1) $E_{train}^{-} = E^{-} + E_{valid}^{+} + E_{test}^{+}$, (2) $E_{valid}^{-} = E^{-} + E_{test}^{+}$, and (3) $E_{test}^{-} = E^{-}$, where $E^{-}$ is the set of all negative pairs (excluding self-loops) in the original graph. Notice that the validation and testing *positive* edges are included in the *negative* training set, and the testing *positive* edges are included in the *negative* validation set. These inclusions simulate the real-world scenario where the testing edges (and the validation edges) are unobserved during validation (training). For large graphs, one can downsample both positive and negative pairs to maintain the same class ratio as the input graph for training and testing.

**Positive masking.** For *unbiased training*, a trick similar to *negative injection* (Zhang & Chen, 2018) in *biased training* is needed to guarantee model generalizability. Specifically, we divide the training positive edges into batches and during the training with each batch $E_b$, we only feed in the residual edges $E - E_b$ to the model. This setting simulates the testing phase, where the model is expected to predict edges without using their own connectivity information. We term this trick *positive masking*.

**Other implementation details.** We add self-loops to the enhanced adjacency matrix to ensure that each node has a valid transition probability distribution that is used in computing Autocovariance. The self-loops are added to all isolated nodes in the training graph for PUBMED, PHOTO, and COMPUTERS, and to all nodes for CORA and CITESEER. Following the postprocessing of the Autocovariance matrix for embedding in Huang et al. (2021), we standardize Gelato similarity scores before computing the loss. For training with the cross entropy loss, we further add a linear layer with the sigmoid activation function to map our prediction score to a probability. We optimize our model with gradient descent via `autograd` in `pytorch` (Paszke et al., 2019). We find that the gradients are sometimes invalid when training our model (especially with the cross entropy loss), and we address this by skipping the parameter updates for batches leading to invalid gradients. Finally, we use $prec@100\%$ on the (unbiased) validation set as the criteria for selecting the best model from all training epochs. The maximum number of epochs for CORA/CITESEER/PUBMED and PHOTO/COMPUTERS are set to be 100 and 250, respectively.

**Experiment environment.** We run our experiments in an `a2-highgpu-1g` node of the Google Cloud Compute Engine. It has one NVIDIA A100 GPU with 40GB HBM2 GPU memory and 12 Intel Xeon Scalable Processor (Cascade Lake) 2nd Generation vCPUs with 85GB memory.

**Reference of baselines.** We list link prediction baselines and their reference repositories we use in our experiments in Table 5. Note that we had to implement the batched training and testing for several baselines as their original implementations do not scale to *unbiased training* and *unbiased testing* without downsampling.

**Number of trainable parameters.** The only trainable component in Gelato is the graph learning MLP, which for `Photo` has 208,130 parameters. By comparison, the best performing GNN-based method, Neo-GNN, has more than twice the number of parameters (455,200).

Table 5: Reference of baseline code repositories.

| Baseline | Repository |
|---|---|
| GAE (Kipf & Welling, 2017) | https://github.com/zfjsail/gae-pytorch |
| SEAL (Zhang & Chen, 2018) | https://github.com/facebookresearch/SEAL_OGB |
| HGCN (Chami et al., 2019) | https://github.com/HazyResearch/hgcn |
| LGCN (Zhang et al., 2021) | https://github.com/ydzhang-stormstout/LGCN/ |
| TLC-GNN (Yan et al., 2021) | https://github.com/pkuyzy/TLC-GNN/ |
| Neo-GNN (Yun et al., 2021) | https://github.com/seongjunyun/Neo-GNNs |
| NBFNet (Zhu et al., 2021) | https://github.com/DeepGraphLearning/NBFNet |
| BScNets (Chen et al., 2022) | https://github.com/BScNets/BScNets |
| WalkPool (Pan et al., 2022) | https://github.com/DaDaCheng/WalkPooling |
| AC (Huang et al., 2021) | https://github.com/zexihuang/random-walk-embedding |

# E RESULTS BASED ON AUC SCORES

As we have argued in Section 2, AUC is not an effective evaluation metric for link prediction (even under *unbiased testing*) as it is biased towards the majority class. In Table 6, we report the AUC scores for different methods under *unbiased testing*. These results, while being consistent with those found in the link prediction literature, disagree with those obtained using the rank-based evaluation metrics under *unbiased testing*.

Table 6: Link prediction performance comparison (mean ± std AUC). AUC results conflict with other evaluation metrics, presenting a misleading view of the model performance for link prediction.

| | | CORA | CITESEER | PUBMED | PHOTO | COMPUTERS |
|---|---|---|---|---|---|---|
| GNN | GAE | 87.30 ± 0.22 | 87.48 ± 0.39 | 94.10 ± 0.22 | 77.59 ± 0.73 | 79.36 ± 0.37 |
| | SEAL | 91.82 ± 1.08 | 90.37 ± 0.91 | *** | 98.85 ± 0.04 | 98.7[*] |
| | HGCN | 92.60 ± 0.29 | 92.39 ± 0.61 | 94.40 ± 0.14 | 96.08 ± 0.08 | 97.86 ± 0.10 |
| | LGCN | 91.60 ± 0.23 | 93.07 ± 0.77 | 95.80 ± 0.03 | 98.36 ± 0.01 | 97.81 ± 0.01 |
| | TLC-GNN | 91.57 ± 0.95 | 91.18 ± 0.78 | OOM | 98.20 ± 0.08 | OOM |
| | Neo-GNN | 91.77 ± 0.84 | 90.25 ± 0.80 | 90.43 ± 1.37 | 98.74 ± 0.55 | 98.34[*] |
| | NBFNet | 86.06 ± 0.59 | 85.10 ± 0.32 | *** | 98.29 ± 0.35 | *** |
| | BScNets | 91.59 ± 0.47 | 89.62 ± 1.05 | 97.48 ± 0.07 | 98.68 ± 0.06 | 98.41 ± 0.05 |
| | WalkPool | 92.33 ± 0.30 | 90.73 ± 0.42 | 98.66[*] | OOM | OOM |
| Topological Heuristics | CN | 71.15 ± 0.00 | 66.91 ± 0.00 | 64.09 ± 0.00 | 96.73 ± 0.00 | 96.15 ± 0.00 |
| | AA | 71.22 ± 0.00 | 66.92 ± 0.00 | 64.09 ± 0.00 | 97.02 ± 0.00 | 96.58 ± 0.00 |
| | RA | 71.22 ± 0.00 | 66.93 ± 0.00 | 64.09 ± 0.00 | 97.20 ± 0.00 | 96.82 ± 0.00 |
| | AC | 75.41 ± 0.00 | 72.41 ± 0.00 | 67.46 ± 0.00 | 98.24 ± 0.00 | 96.86 ± 0.00 |
| Attributes + Topology | MLP | 85.43 ± 0.36 | 87.89 ± 2.05 | 87.89 ± 2.05 | 91.24 ± 1.61 | 88.84 ± 2.58 |
| | Cos | 79.06 ± 0.00 | 89.86 ± 0.00 | 89.14 ± 0.00 | 71.46 ± 0.00 | 71.68 ± 0.00 |
| | MLP+AC | 79.77 ± 0.03 | 82.26 ± 0.29 | 66.31 ± 0.74 | 98.02 ± 0.05 | 96.69 ± 0.05 |
| | Cos+AC | 86.34 ± 0.00 | 89.29 ± 0.00 | 75.56 ± 0.00 | 97.28 ± 0.00 | 96.26 ± 0.00 |
| | MLP+Cos+AC | 86.90 ± 0.14 | 89.42 ± 0.09 | 75.78 ± 0.27 | 97.27 ± 0.01 | 96.24 ± 0.01 |
| Gelato | | 83.12 ± 0.06 | 88.38 ± 0.02 | 64.93 ± 0.06 | 98.01 ± 0.03 | 96.72 ± 0.04 |

[*] Run only once as each run takes ~100 hrs;  *** Each run takes >1000 hrs;  OOM: Out Of Memory.

## F    VISUALIZING LINK PREDICTION PROCESSES

In this section, we build upon the discussion in Section 4.3 and present a more detailed analysis of the improvements of Gelato over Autocovariance based on the input graph and comparisons with the link prediction process of the best GNN-based baseline (Neo-GNN) in Figure 10.

Figure 10a shows the input adjacency matrix for the subgraph of PHOTO containing the top 160 nodes belonging to the first class sorted in decreasing order of their (within-class) degree. Figure 10b illustrates the enhanced adjacency matrix learned by Gelato's MLP. Comparing it with the Euclidean distance between node attributes in Figure 10c, we see that our enhanced adjacency matrix effectively incorporates attribute information. More specifically, we notice the down-weighting of the edges connecting the high-degree nodes with larger attribute distances (matrix entries 0-40 and especially 0-10) and the up-weighting of those connecting medium-degree nodes with smaller attribute distances (40-80). In Figure 10d and Figure 10e, we see the corresponding AC scores based on the input and the enhanced adjacency matrix (Gelato). Since AC captures the degree distribution of nodes (Huang et al., 2021), the vanilla AC scores greatly favor high-degree nodes (0-40). By comparison, thanks to the down-weighting, Gelato assigns relatively lower scores to edges connecting them to low-degree nodes (60-160), while still capturing the edge density between high-degree nodes (0-40). The immediate result of this is the significantly improved precision as shown in Figure 10h compared to Figure 10g. Gelato covers as many positive edges in the high-degree region as AC while making far fewer wrong predictions for connections involving low-degree nodes.

The prediction probabilities and predicted edges for Neo-GNN are shown in Figure 10f and Figure 10i, respectively. Note that while it predicts edges connecting high-degree node pairs (0-40) with high probability, similar values are assigned to many low-degree pairs (80-160) as well. Most of those predictions are wrong, both in the low-degree region of Figure 10i and also in other low-degree parts of the graph that are not shown here. This analysis supports our claim that Gelato is more effective at combining node attributes and the graph topology, enabling state-of-the-art link prediction.

## G    ABLATION STUDY

We have demonstrated the superiority of Gelato over its individual components and two-stage approaches in Table 2 and analyzed the effect of losses and training settings in Section 4.4. Here, we collect the results with the same hyperparameter setting as Gelato and present a comprehensive ablation study in Table 7. Specifically, Gelato − MLP (AC) represents Gelato without the MLP (Autocovariance) component, i.e., only using Autocovariance (MLP) for link prediction. Gelato − NP (UT) replaces the proposed N-pair loss (*unbiased training*) with the cross entropy loss (*biased training*) applied by the baselines, and Gelato − NP+UT replaces both the loss and the training setting.

Table 7: Results of the ablation study based on AP scores. Each component of Gelato plays an important role in enabling state-of-the-art link prediction performance.

|  | CORA | CITESEER | PUBMED | PHOTO | COMPUTERS |
|---|---|---|---|---|---|
| Gelato − MLP | 2.43 ± 0.00 | 2.65 ± 0.00 | 2.50 ± 0.00 | 16.63 ± 0.00 | 11.64 ± 0.00 |
| Gelato − AC | 1.94 ± 0.18 | 3.91 ± 0.37 | 0.83 ± 0.05 | 7.45 ± 0.44 | 4.09 ± 0.16 |
| Gelato − NP+UT | 2.98 ± 0.20 | 1.96 ± 0.11 | 2.35 ± 0.24 | 14.87 ± 1.41 | 9.77 ± 2.67 |
| Gelato − NP | 1.96 ± 0.01 | 1.77 ± 0.20 | 2.32 ± 0.16 | 19.63 ± 0.38 | 9.84 ± 4.42 |
| Gelato − UT | 3.07 ± 0.01 | 1.95 ± 0.05 | 2.52 ± 0.09 | 23.66 ± 1.01 | 11.59 ± 0.35 |
| Gelato | **3.90 ± 0.03** | **4.55 ± 0.02** | **2.88 ± 0.09** | **25.68 ± 0.53** | **18.77 ± 0.19** |

We observe that removing either MLP or Autocovariance leads to inferior performance, as the corresponding attribute or topology information would be missing. Further, to address the class imbalance problem of link prediction, both the N-pair loss and *unbiased training* are crucial for effective training of Gelato.

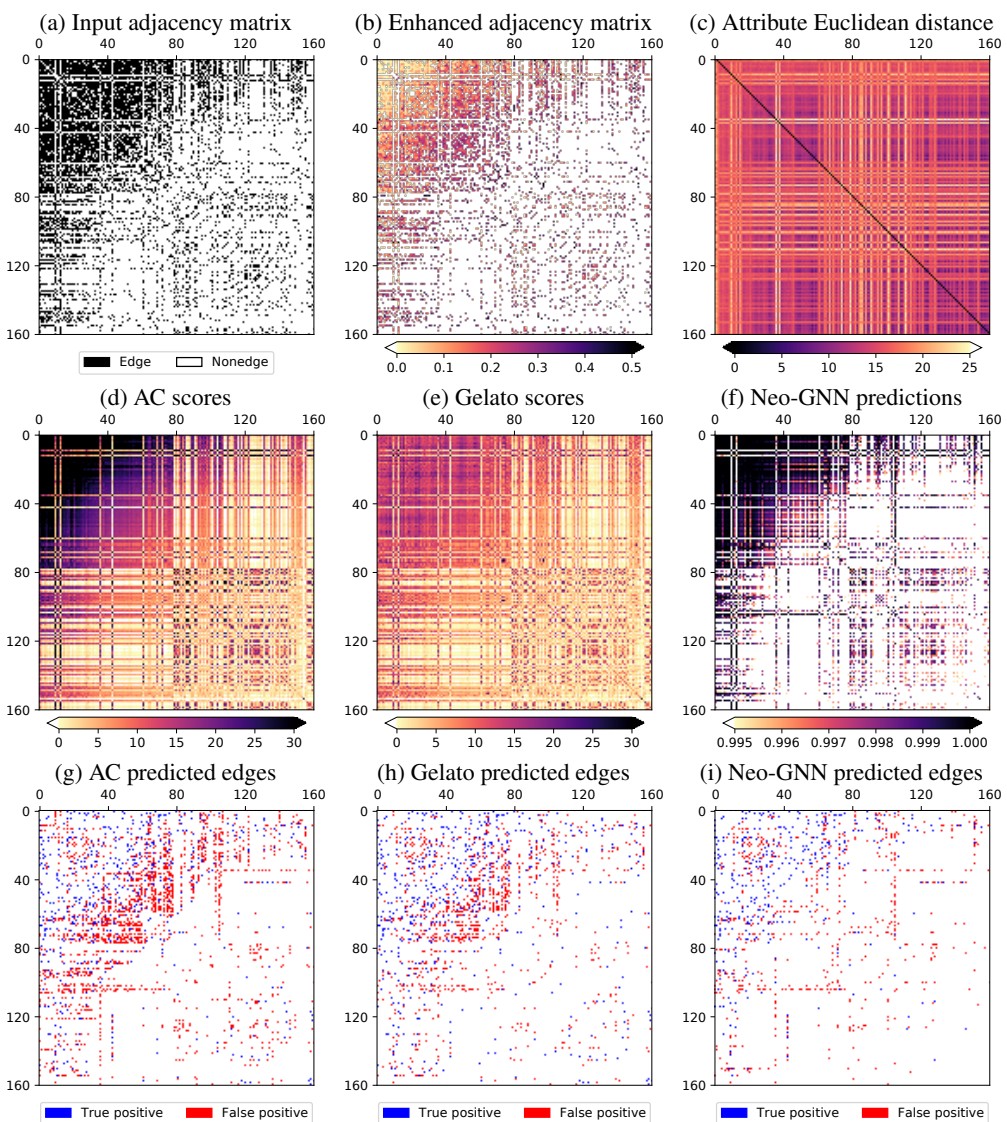

Figure 10: Illustration of the link prediction process of Gelato, AC, and the best GNN-based approach, Neo-GNN, on a subgraph of PHOTO. Gelato effectively incorporates node attributes into the graph structure and leverages topological heuristics to enable state-of-the-art link prediction.

# H    BASELINE RESULTS WITH UNBIASED TRAINING

Table 8 summarizes the link prediction performance in terms of mean and standard deviation of AP for Gelato and the baselines using *unbiased training* without downsampling and the cross entropy loss, and Figure 11 and Figure 12 show results based on $prec@k$ and $hits@k$ for varying $k$ values.

Table 8: Link prediction performance comparison (mean ± std AP) with supervised link prediction methods using *unbiased training*. While we observe noticeable improvement for some baselines (e.g., BScNets), Gelato still consistently and significantly outperform the baselines.

|  |  | CORA | CITESEER | PUBMED | PHOTO | COMPUTERS |
|---|---|---|---|---|---|---|
| GNN | GAE | 0.33 ± 0.21 | 0.69 ± 0.18 | 0.63[*] | 1.36 ± 3.38 | 7.91[*] |
|  | SEAL | 2.24[*] | 1.11[*] | *** | *** | *** |
|  | HGCN | 0.54 ± 0.23 | 1.02 ± 0.05 | 0.41[*] | 3.27 ± 2.97 | 2.60[*] |
|  | LGCN | 1.53 ± 0.08 | 1.45 ± 0.09 | 0.55[*] | 2.90 ± 0.26 | 1.13[*] |
|  | TLC-GNN | 0.68 ± 0.16 | 0.61 ± 0.19 | OOM | 2.95 ± 1.50 | OOM |
|  | Neo-GNN | 2.76 ± 0.36 | 1.80 ± 0.22 | *** | *** | *** |
|  | NBFNet | *** | *** | *** | *** | *** |
|  | BScNets | 1.13 ± 0.25 | 1.27 ± 0.20 | 0.47[*] | 8.54 ± 1.55 | 4.40[*] |
|  | WalkPool | *** | *** | *** | OOM | OOM |
| Topological Heuristics | CN | 1.10 ± 0.00 | 0.74 ± 0.00 | 0.36 ± 0.00 | 7.73 ± 0.00 | 5.09 ± 0.00 |
|  | AA | 2.07 ± 0.00 | 1.24 ± 0.00 | 0.45 ± 0.00 | 9.67 ± 0.00 | 6.52 ± 0.00 |
|  | RA | 2.02 ± 0.00 | 1.19 ± 0.00 | 0.33 ± 0.00 | 10.77 ± 0.00 | 7.71 ± 0.00 |
|  | AC | 2.43 ± 0.00 | 2.65 ± 0.00 | 2.50 ± 0.00 | 16.63 ± 0.00 | 11.64 ± 0.00 |
| Attributes + Topology | MLP | 0.22 ± 0.27 | 1.17 ± 0.63 | 0.44 ± 0.12 | 1.15 ± 0.40 | 1.19 ± 0.46 |
|  | Cos | 0.42 ± 0.00 | 1.89 ± 0.00 | 0.07 ± 0.00 | 0.11 ± 0.00 | 0.07 ± 0.00 |
|  | MLP+AC | 0.63 ± 0.32 | 1.00 ± 0.43 | 1.17 ± 0.44 | 11.88 ± 0.83 | 8.73 ± 0.45 |
|  | Cos+AC | 3.60 ± 0.00 | 4.46 ± 0.00 | 0.51 ± 0.00 | 10.01 ± 0.00 | 5.20 ± 0.00 |
|  | MLP+Cos+AC | 3.80 ± 0.01 | 3.94 ± 0.03 | 0.77 ± 0.01 | 12.80 ± 0.03 | 7.57 ± 0.02 |
| Gelato |  | **3.90 ± 0.03** | **4.55 ± 0.02** | **2.88 ± 0.09** | **25.68 ± 0.53** | **18.77 ± 0.19** |

[*] Run only once as each run takes ~100 hrs;    *** Each run takes >1000 hrs;    OOM: Out Of Memory.

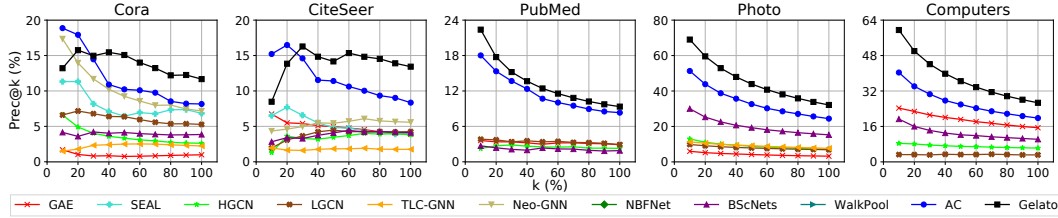

Figure 11: Link prediction performance in terms of $prec@k$ (in percentage) for varying values of $k$ with baselines using *unbiased training*. While we observe noticeable improvement for some baselines (e.g., BScNets), Gelato still consistently and significantly outperform the baselines.

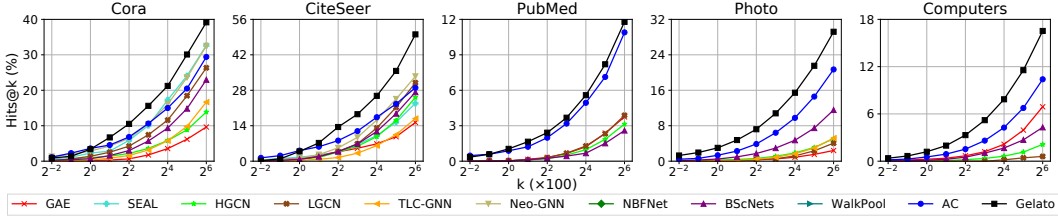

Figure 12: Link prediction performance in terms of $hits@k$ (in percentage) for varying values of $k$ with baselines using *unbiased training*. While we observe noticeable improvement for some baselines (e.g., BScNets), Gelato still consistently and significantly outperform the baselines.

We first note that *unbiased training* without downsampling brings serious scalability challenges to most GNN-based approaches, making scaling up to larger datasets intractable. For the scalable baselines, *unbiased training* leads to marginal (e.g., Neo-GNN) to significant (e.g., BScNets) gains in performance. However, all of them still underperform the topological heuristic, Autocovariance, in most cases, and achieve much worse performance compared to Gelato. This supports our claim that the topology-centric graph learning mechanism in Gelato is more effective than the attribute-centric message-passing in GNNs for link prediction, leading to state-of-the-art performance.

As for the GNN-based baselines using both *unbiased training* and our proposed N-pair loss, we have attempted to modify the training functions of the reference repositories of the baselines and managed to train SEAL, LGCN, Neo-GNN, and BScNet. However, despite our best efforts, we are unable to obtain better link prediction performance using the N-pair loss. We defer the further investigation of the incompatibility of our loss and the baselines to future research. On the other hand, we have observed significantly better performance for MLP with the N-pair loss compared to the cross entropy loss under *unbiased training*. This can be seen by comparing the MLP performance in Table 8 here and the Gelato − AC performance in Table 7 in the ablation study. The improvement shows the potential benefit of applying our training setting and loss to other supervised link prediction methods.

We also show the training time comparison between different supervised link prediction methods using *unbiased training* without downsampling in Table 9. Gelato is the second fastest model, only slower than the vanilla MLP. This further demonstrates that Gelato is a more efficient alternative compared to GNNs.

Table 9: Training time comparison between supervised link prediction methods for PHOTO under *unbiased training*. Gelato, while achieving the best performance, is also the second most efficient method in terms of total training time, slower only than the vanilla MLP.

|  | GAE | SEAL | HGCN | LGCN | TLC-GNN | Neo-GNN | NBFNet | BScNets | MLP | Gelato |
|---|---|---|---|---|---|---|---|---|---|---|
| Training | 6,361 | >450,000 | 1,668 | 1,401 | 53,304 | >450,000 | >450,000 | 2,323 | 744 | 1,285 |

