# OpenReview forum: "Link Prediction without Graph Neural Networks"
_ICLR.cc/2023/Conference — Submitted to ICLR 2023_

### Official Review · Reviewer_maA8 · 2022-10-25

**Confidence:** 5
**Correctness:** 3
**Technical Novelty And Significance:** 2
**Empirical Novelty And Significance:** 3
**Recommendation:** 1

**Clarity, Quality, Novelty And Reproducibility:**

Clarity:
 - Good

Quality
 - Empirical evaluation datasets are poor, models, however, are well chosen

Novelty
 - Low to Medium, the technical innovation is not super significant.

Reproducibility
- Good



**Strength And Weaknesses:**

Weaknesses:
1. A low-effort re-submission!!
2. Doesn't cite and compare to the relevant Knowledge graph link prediction literature, which for a long time now uses unbiased testing (i.e. the idea is not new) as well as failing to compare their loss function to negative sampling from the KG community.
3. The benchmark datasets are simply not interesting anymore and are not sufficient to validate the empirical performance of their method, they should incorporate the homogenous graph link prediction datasets from OGB at least.

Strengths:
1. Method is efficient.
2. Clear writing.
3. Rightly criticises biased testing.
4. Models that are compared against are representative.

**Summary Of The Paper:**

This paper is a re-submission. I have previously reviewed and the authors made only minor changes without incorporating the feedback from their prior submission.

The paper rightly critises biased testing in homogenous graph link prediction and proposes to combine topological features with MLPs for this problem.

**Summary Of The Review:**

A re-submission that doesn't adress the weaknessess from the last round of feedback (see Weaknessess section). Clear reject.

---

> ### Author Response · Authors · 2022-11-13
> **Response to Reviewer maA8**
>
> Thank you for taking the time to review our submission again and providing constructive comments! Please find our responses below:
>
> > A low-effort re-submission!!
>
> Based on your previous comments, we have rerun our experiments and incorporated results based on Hits@k from knowledge graph community and Average Precision. We have also added a detailed sensitivity analysis as well as multiple references to the knowledge graph literature in the evaluation metrics and loss functions.
>
> > Doesn't cite and compare to the relevant Knowledge graph link prediction literature, which for a long time now uses unbiased testing (i.e. the idea is not new) as well as failing to compare their loss function to negative sampling from the KG community.
>
> The following references were added to the current submission to acknowledge the contributions of the knowledge graph community:
> * Evaluation metrics (Page 3):
>     * Translating embeddings for modeling multi-relational data. NeurIPS 2013.
>     * Embedding entities and relations for learning and inference in knowledge bases. ICLR 2015.
>     * Rotate: Knowledge graph embedding by relational rotation in complex space. ICLR 2016.
> * Loss functions and negative sampling techniques (Page 5):
>     * Adversarial learning for knowledge graph embeddings. ACL 2018.
>     * Incorporating gan for negative sampling in knowledge representation learning. AAAI 2018.
>
> While the evaluation metrics, the loss functions, and the negative sampling techniques have been applied in the knowledge graph community, we have not seen their applications in homogenous graph link prediction in even the most recent papers (e.g., BScNet, WalkPool). Part of the goal of this paper is to bring those ideas from the knowledge graph community to benefit the link prediction community.
>
> > The benchmark datasets are simply not interesting anymore and are not sufficient to validate the empirical performance of their method, they should incorporate the homogenous graph link prediction datasets from OGB at least.
>
> We follow existing approaches in our dataset selection. Specifically, Cora, CiteSeer, and PubMed are used in LGCN (WebConf’21), NBFNet (NeurIPS’21), TLC-GNN (ICML’21), BScNet (AAAI’22), WalkPool (ICLR’22), CFLP (ICML’22), and Photo and Computers are used in LGCN (WebConf’21) and TLC-GNN (ICML’21). Notably, we are even unable to scale the WalkPool method appearing in ICLR’22 to all datasets used here for unbiased testing, along with other recent approaches (SEAL, TLC-GNN, Neo-GNN, NBFNet). Moreover, our results show that even approaches that do scale to larger datasets, such as those in OGB, are significantly outperformed by Autocovariance (AC), which is a much simpler topology-based heuristic that does not account for node features (see Table 2). This is evidence that improving accuracy is a bigger challenge than improving scalability for recent link prediction methods.

---

> > ### Comment · Reviewer_maA8 · 2022-12-01
> > **Benchmark datasets change over time**
> >
> > While prior work may have used these benchmark datasets, they have proven to be too easy at this stage and not informative for empirical evaluation anymore. With time the benchmark datasets change, so prior work having used it is not surprising, given the easily accessable more current benchmark datasets, this remains a blocker for acceptance.

---

> > > ### Author Response · Authors · 2022-12-02
> > > **Task difficulty and dataset age**
> > >
> > > We thank the reviewer for giving us another chance to address their concerns. Please find our responses below:
> > >
> > > > While prior work may have used these benchmark datasets, they have proven to be too easy at this stage and not informative for empirical evaluation anymore.
> > >
> > > We agree with the reviewer that based on the evaluation scheme in the current literature---AUC/AP under biased testing---most link prediction approaches can easily achieve scores of often higher than 90%, making it hard to compare different methods. However, as we argue in our paper, biased testing provides an overly optimistic performance evaluation of a link prediction model in real-world scenarios. And with unbiased testing, even the best GNN-based method, Neo-GNN, achieves only ~10% AP at best among the aforementioned datasets, which shows how __challenging__ link prediction is in those datasets. In addition, unlike previous papers which report marginal improvements that are called into question, our proposed method __more than doubles the AP scores__ compared to existing approaches across the datasets.
> > >
> > > > With time the benchmark datasets change, so prior work having used it is not surprising, given the easily accessable more current benchmark datasets, this remains a blocker for acceptance.
> > >
> > > The reviewer has not provided any reason why our results are not representative, except that our datasets might have been created before the publication of the OGB paper. If the reviewers collectively agree that dataset age itself is a __blocker for acceptance__, we strongly disagree with that.
> > >
> > > A key contribution of our paper is exactly to criticize biased training and testing, as applied by the few GNN-based link prediction papers using OGB datasets and by OGB’s evaluation procedure as well. To the best of our knowledge, there are no GNN-based link prediction papers using OGB with unbiased training or testing due to obvious scalability challenges. In case the reviewer claims that scalability is a __blocker for acceptance__, we also strongly disagree. Most applications of graph ML, especially those outside big tech companies, such as social good and biology, involve small datasets. Moreover, focusing on scalability only favors research groups with easy access to multiple GPUs with large amounts of memory, which are quite expensive and currently inaccessible to us. We refer anyone interested in a more comprehensive discussion on these issues to [1].
> > >
> > > [1] Bommasani et al. On the opportunities and risks of foundation models. arXiv preprint arXiv:2108.07258, 2021.

---

### Official Review · Reviewer_SSk7 · 2022-10-25

**Confidence:** 5
**Correctness:** 3
**Technical Novelty And Significance:** 2
**Empirical Novelty And Significance:** 2
**Recommendation:** 6

**Clarity, Quality, Novelty And Reproducibility:**

The paper is clear and somehow novel. There is a separate section to describe the resources for Reproducibility.

**Strength And Weaknesses:**

Pros: the authors scrutinize the training and evaluation of supervised link prediction methods and identify their limitations in handling class imbalance; the authors propose a simple, effective, and efficient framework to combine topological and attribute information for link prediction without using GNNs; and they introduce an N-pair link prediction loss combined with an unbiased set of training edges that we show to be more effective at addressing the class imbalance.

Cons: The authors only use some small datasets for evaluation, on which there do not exist standard splits for different methods to be evaluated in a standard manner. The authors are suggested to test their method on larger datasets such as OGBL ones to comprehensively evaluate their method.

**Summary Of The Paper:**

The authors first the important limitations in how GNN-based link prediction methods handle the intrinsic class imbalance of the problem—due to the graph sparsity—in their training and evaluation. Moreover, the authors propose Gelato, a novel topology-centric framework that
applies a topological heuristic to a graph enhanced by attribute information via graph learning. Gelato is trained end-to-end with an N-pair loss on an unbiased training set to address the class imbalance.

**Summary Of The Review:**

Please refer to the above review comments for improvements.

---

> ### Author Response · Authors · 2022-11-13
> **Response to Reviewer SSk7**
>
> Thank you for taking the time to review our submission and providing constructive comments! Please find our responses below:
>
> > The authors only use some small datasets for evaluation, on which there do not exist standard splits for different methods to be evaluated in a standard manner. The authors are suggested to test their method on larger datasets such as OGBL ones to comprehensively evaluate their method.
>
> * We follow existing approaches in our dataset selection. Specifically, Cora, CiteSeer, and PubMed are used in LGCN (WebConf’21), NBFNet (NeurIPS’21), TLC-GNN (ICML’21), BScNet (AAAI’22), WalkPool (ICLR’22), CFLP (ICML’22), and Photo and Computers are used in LGCN (WebConf’21) and TLC-GNN (ICML’21). Notably, we are even unable to scale the WalkPool method appearing in ICLR’22 to all datasets used here for unbiased testing, along with other recent approaches (SEAL, TLC-GNN, Neo-GNN, NBFNet). Moreover, our results show that even approaches that do scale to larger datasets, such as those in OGB, are significantly outperformed by Autocovariance (AC), which is a much simpler topology-based heuristic that does not account for node features (see Table 2). This is evidence that improving accuracy is a bigger challenge than improving scalability for recent link prediction methods.
>
> * The split for positive edges follows the same procedure as in existing approaches (GAE, SEAL, HGCN, LGCN, BScNet, WalkPool). For negative edges, as we have argued in Section 2, we adopt unbiased testing and do not downsample the negative pairs. For reference, we also reproduce the AUC results for existing approaches in Table 6 in Appendix E.

---

### Official Review · Reviewer_L8KX · 2022-10-27

**Confidence:** 4
**Correctness:** 3
**Technical Novelty And Significance:** 2
**Empirical Novelty And Significance:** 3
**Recommendation:** 3

**Clarity, Quality, Novelty And Reproducibility:**

The approach presented in clear in details. The level of novelty is somewhat limited and the proposed approach could be further improved/refined. The authors made publicly available their code which allowed for reproducibility checking.

**Strength And Weaknesses:**

Here below are some associated strengths of the paper :

1) Even though graph augmentation based techniques previously exist in the literature, the authors present an interesting perspective of solving the link prediction problem without using GNN explicitly.

2) The authors highlight and make an effort to solve the class imbalance problem in graph datasets which hampers model performance significantly.

3) The approach proposed is a lightweight one which can easily be incorporated and would potentially help model performance in the link prediction problem.

4) The empirical results demonstrate the efficacy of the approach effectively.

Here below are some associated weaknesses of the paper :

1) While there is some merit in the approach presented and the expectation from the reviewer is that this current work might usher in a new field of work, in general the level of novelty of the current approach is incremental in nature and requires more refinement. For example, graph augmentation/enrichment is known in literature as well as metric learning based loss functions as well as the auto covariance based similarity matrix which has demonstrated excellent performance for link prediction based tasks.

2) The approach felt pretty heuristic. In particular the approach is strongly dependent on a similarity metric/function and an associated threshold. The paper does not contain any discussion as to how to choose this similarity metric/function and the associated threshold in general. The authors themselves choose cosine similarity but does this work well in all scenarios. The current work requires more discussion in general.

3) The current work should have included more baselines to compare against.

4) The current work does not list any limitations and/or potential directions for improvement which would have significantly added to the value associated with the work. As mentioned earlier, the current work requires more discussion to be included in general.

**Summary Of The Paper:**

In this work, the authors propose an interesting and novel technique towards solving the problem of link prediction in graphs without using GNN based approach which happens to be the predominant methodology at present. The authors also demonstrate training their model using an N-pair loss (akin to metric learning based loss) to address class imbalance in real-world graph datasets. The authors demonstrate the efficacy of their approach via empirical results.

**Summary Of The Review:**

The authors present an interesting and lightweight approach towards solving the problem of link prediction. The main issues with the current work is the limited level of novelty and the heuristic aspects of the work which require more discussion and/or more work.

---

> ### Author Response · Authors · 2022-11-13
> **Response to Reviewer L8KX**
>
> Thank you for taking the time to review our submission and providing constructive comments! Please find our responses below:
>
> > While there is some merit in the approach presented and the expectation from the reviewer is that this current work might usher in a new field of work, in general the level of novelty of the current approach is incremental in nature and requires more refinement. For example, graph augmentation/enrichment is known in literature as well as metric learning based loss functions as well as the auto covariance based similarity matrix which has demonstrated excellent performance for link prediction based tasks.
>
> We acknowledge the contributions of prior work in graph learning, ranking losses, and topological heuristics. However, the goal of this paper is to investigate whether these components---none of which have been applied in attributed graph link prediction---can achieve similar or even better performance than the increasingly popular GNN architectures, when combined in a novel way. Our work provides an affirmative answer to this, and shows that the topology-centric graph learning paradigm is more suitable for the link prediction task compared to the message-passing mechanism in GNNs. A similar investigation that compares GNNs with simple graph-based solutions for node classification has appeared in ICLR (Combining label propagation and simple models out-performs graph neural networks, ICLR 2021).
>
> > The approach felt pretty heuristic. In particular the approach is strongly dependent on a similarity metric/function and an associated threshold. The paper does not contain any discussion as to how to choose this similarity metric/function and the associated threshold in general. The authors themselves choose cosine similarity but does this work well in all scenarios. The current work requires more discussion in general.
>
> We have included a detailed discussion on our hyperparameter choices in Appendix C, including the criteria for which we select the threshold $\eta$. Specifically, the motivation for adding edges based on the cosine similarity of node features for Cora and CiteSeer is based on our observation that this alone leads to improved performance (see Cos + AC vs AC results in Table 2), while for all other datasets we simply drop the untrained similarity component. The key contribution of the paper is the general framework, and we agree that more discussion on the choices of individual components in the framework can be added in the future work.
>
> > The current work should have included more baselines to compare against.
>
> We have included __all__ state-of-the-art GNN-based approaches in the past two years up to ICLR’22 when we prepared this manuscript, along with pioneering work, topological heuristics, and two-stage approaches. These baselines allow us to conduct a more comprehensive experimental study compared to other ICLR’23 submissions (e.g., https://openreview.net/forum?id=m1oqEOAozQU). We will include more recent baselines when their codes become available.
>
> > The current work does not list any limitations and/or potential directions for improvement which would have significantly added to the value associated with the work. As mentioned earlier, the current work requires more discussion to be included in general.
>
> We fully agree with the reviewer that a detailed discussion on the choices of individual components in our framework is a meaningful future direction for this paper. The goal of this paper is to demonstrate the effectiveness of the proposed topology-centric framework, and we have shown that it leads to significantly better performance than GNNs without an extensive or exhaustive search of its components. Another future direction we are considering is to provide theoretical justification (e.g. in terms of expressive power) for our strong empirical results. We will add a paragraph in the revised paper to discuss these future directions.

---

### Official Review · Reviewer_p64G · 2022-10-27

**Confidence:** 3
**Correctness:** 3
**Technical Novelty And Significance:** 2
**Empirical Novelty And Significance:** 2
**Recommendation:** 3

**Clarity, Quality, Novelty And Reproducibility:**

The paper is clearly written and easy to follow. The technical contribution could be improved and several claims need further justification. The authors provide anonymous code but the reviewer does not check the code.

**Strength And Weaknesses:**

Strengths:
+ This paper is easy to read and follow.
+ The proposed strategy significantly outperforms existing benchmarks and especially improves efficiency.

Weaknesses:
- Several statements may be too strong to claim. For example: in Page 2 the authors claim existing approaches "highly overestimate" the ratio of positive pairs. I feels skeptical about this claim, as the negative pairs are randomly drawn rather than fixed. It is not clear to me why they are biased towards positive pairs. Similarly, I am not clear about why prior approaches need to construct a biased test dataset.
- I in person believe there is a gap between the motivation and the model design. The only connection I can see is the objective Eq. (16) uses a full set of negative links, where prior approaches can do this as well.
- Experimental results are not fully convincing. On several experiments (Cora, Citeseer, Pubmed) the improvements seem marginal. The overall dataset sizes are small. It seems that the model is very sensitive to the parameter $\eta$. More explanations are needed. Also, more head-to-head ablation studies are needed. For example, I am curious whether the graph structure learning component (Attributes+Topology) can be used with conventional training schemes.

**Summary Of The Paper:**

This paper presents a model for link prediction without graph neural networks. The authors firstly examine existing benchmarks and argue they are biased towards positive links. Then, they propose an effective framework with graph structure learning with a topology heuristic. A series experiments show the superior performance of the proposed approach.

**Summary Of The Review:**

I think this work provides an important contribution to "graph-less" neural networks for link prediction, but I feel like the motivation of this work needs further elaboration. Some claims on existing "biased" models are too strong.

---

> ### Author Response · Authors · 2022-11-13
> **Response to Reviewer p64G**
>
> Thank you for taking the time to review our submission and providing constructive comments! Please find our responses below:
>
> > Several statements may be too strong to claim. For example: in Page 2 the authors claim existing approaches "highly overestimate" the ratio of positive pairs. I feels skeptical about this claim, as the negative pairs are randomly drawn rather than fixed. It is not clear to me why they are biased towards positive pairs. Similarly, I am not clear about why prior approaches need to construct a biased test dataset.
>
> With $m=O(n)$ positive edges, existing approaches randomly draw $m$ samples from $O(n^2)$ negative pairs for their evaluation. Ranking $m$ positive edges against $m$ negative pairs instead of $O(n^2)$ is a significantly easier problem, and evaluation based on this biased test set presents an overly optimistic view of the model performance. We refer the reviewer to the following paper and our illustrative example in Appendix A for more details on biased vs unbiased testing in imbalance datasets.
>
> The Relationship Between Precision-Recall and ROC Curves. Jesse Davis, Mark Goadrich. ICML 2006.
>
> > I in person believe there is a gap between the motivation and the model design. The only connection I can see is the objective Eq. (16) uses a full set of negative links, where prior approaches can do this as well.
>
> * The goal of this paper is to investigate both the training and evaluation of GNN-based link prediction methods and __their ability to learn effective topological heuristics directly from data__. The proposed framework, Gelato, presents an alternative way to incorporate node attributes and directly leverage topological heuristics, which we have found to lead to significantly better performance. We believe this paradigm shift could help us better understand the weaknesses of existing GNN-based approaches and build better methods that draw insights from this simpler graph-based solution.
>
> * We have provided detailed experimental results that combine prior approaches with the proposed unbiased training scheme in Appendix H. While we have observed noticeable improvement for some approaches (e.g., BScNets), Gelato still consistently and significantly outperforms all prior approaches.
>
> > Experimental results are not fully convincing. On several experiments (Cora, Citeseer, Pubmed) the improvements seem marginal. The overall dataset sizes are small.
>
> * Compared to the best GNN-based method, Neo-GNN, Gelato achieves __137%__ better performance in the aforementioned datasets in terms of AP, as well as __49%__ better over Autocovariance. We believe these gaps are significant to demonstrate the advantages of Gelato.
>
> * For our dataset selection choices, Cora, CiteSeer, and PubMed are used in LGCN (WebConf’21), NBFNet (NeurIPS’21), TLC-GNN (ICML’21), BScNet (AAAI’22), WalkPool (ICLR’22), CFLP (ICML’22), and Photo and Computers are used in LGCN (WebConf’21) and TLC-GNN (ICML’21). Notably, we are even unable to scale the WalkPool method appearing in ICLR’22 to all datasets used here for unbiased testing, along with other recent approaches (SEAL, TLC-GNN, Neo-GNN, NBFNet). Moreover, our results show that even approaches that do scale to larger datasets, such as those in OGB, are significantly outperformed by Autocovariance (AC), which is a much simpler topology-based heuristic that does not account for node features (see Table 2). This is evidence that improving accuracy is a bigger challenge than improving scalability for recent link prediction methods.
>
> > It seems that the model is very sensitive to the parameter $\eta$.
>
> We have provided a detailed discussion on the hyperparameter choices in Appendix C. For most datasets we have found $\eta = 0.0$ (i.e., no added edges) leads to the best performance. The reason for adding feature-based edges for Cora and CiteSeer is based on our observation that this alone leads to improved performance (see Cos + AC vs AC results in Table 2). Also, based on our experiments, all hyperparameters can be easily tuned based on simple hyperparameter search techniques, such as line search, using a small validation set.
>
> > More explanations are needed. Also, more head-to-head ablation studies are needed. For example, I am curious whether the graph structure learning component (Attributes+Topology) can be used with conventional training schemes.
>
> We have provided ablation studies for both Gelato (Appendix G) and the baselines (Appendix H). Specifically, the results for our end-to-end framework Gelato combined with biased training and/or cross entropy loss can be found in Table 7 in Appendix G, and explanations of the effect of training schemes and loss functions are presented in Section 4.4. The results for the two-stage approach (Attributes+Topology) combined with biased training and cross entropy loss can be found in Table 8 in Appendix H. We are more than happy to add any other specific ablation study results based on the reviewer’s request.

---

### Decision · Program_Chairs · 2023-01-20

**Decision:**

Reject

**Justification For Why Not Higher Score:**

This paper has unconvincing experiments. The reasons are stated in the meta review. It is questionable why a simple hacked heuristic can outperform sota GNNs.

**Justification For Why Not Lower Score:**

N/A

**Metareview: Summary, Strengths And Weaknesses:**

This paper studies link prediction without graph neural networks. It firstly examines existing benchmarks and argues they are biased towards positive links. A simple topology heuristic-based learning model rained on unbiased training data combined with N-pair loss is proposed to address the issue, which outperforms advanced GNNs under the unbiased training/testing setting. However, as pointed out by many reviewers, one severe limitation is that the paper only uses outdated small datasets for evaluation. The authors refused to perform experiments on ogb-level datasets with reasons that do not make much sense to me. Small datasets are easy to hack. Under a different training/testing setting, existing strong GNN baselines may need some significant tuning to work well. Only adopting their original hyperparameters make the comparison not solid, which is another concern.